# Secretases Related to Amyloid Precursor Protein Processing

**DOI:** 10.3390/membranes11120983

**Published:** 2021-12-15

**Authors:** Xiaoling Liu, Yan Liu, Shangrong Ji

**Affiliations:** 1Key Laboratory of Cell Activities and Stress Adaptations, Ministry of Education, School of Life Sciences, Lanzhou University, South Tianshui Road, Lanzhou 730030, China; liuxl19@lzu.edu.cn; 2Key Laboratory of Environment and Genes Related to Diseases, Ministry of Education, School of Basic Medical Sciences, Xi’an Jiaotong University, West Yanta Road, Xi’an 710061, China; liuyan111o@163.com

**Keywords:** Alzheimer’s disease, amyloid precursor protein, secretase, APP processing

## Abstract

Alzheimer’s disease (AD) is a common neurodegenerative disease whose prevalence increases with age. An increasing number of findings suggest that abnormalities in the metabolism of amyloid precursor protein (APP), a single transmembrane aspartic protein that is cleaved by β- and γ-secretases to produce β-amyloid protein (Aβ), are a major pathological feature of AD. In recent years, a large number of studies have been conducted on the APP processing pathways and the role of secretion. This paper provides a summary of the involvement of secretases in the processing of APP and the potential drug targets that could provide new directions for AD therapy.

## 1. Introduction

Alzheimer’s disease (AD) is the most common type of dementia, with approximately 60% of clinical cases of dementia being AD [1]. The main risk factor for AD is age, with the incidence doubling every five years after age 65. In addition, a small number of AD cases are inherited in an autosomal-dominant manner, and carriers may develop this disease after approximately 40 years of age. Clinically, AD is mainly characterized by progressive memory impairment, cognitive dysfunction and language impairment [2]. There are many hypotheses about the pathogenesis of AD, and the most widely accepted is the amyloid cascade hypothesis [3,4,5]. Amyloid precursor protein (APP) is proteolyzed by β-secretase and γ-secretase to produce the β-amyloid (Aβ) protein. When the normal expression and processing of APP are affected, the balance between Aβ production and clearance gradually changes. The accumulation of Aβ triggers a series of cascade reactions, including tau protein phosphorylation, neurotransmitter loss, and neuroinflammatory reactions, eventually leading to neuronal dysfunction, death, inflammatory plaque formation, neurofibrillary tangle formation and other pathological phenomena. Although the etiology and pathogenesis of AD are not well understood, the polymerization and deposition of Aβ, which is produced by abnormal metabolism of APP, is considered to be an important cause of AD [6]. α-, β- and γ-Secretases play a crucial role in the metabolism of APP.

AD is classified by the age of onset. If the age of onset is less than 65 years old, the disease is classified as familial AD (FAD), and if the age of onset is more than 65 years old, the disease is regarded as sporadic AD. FAD accounts for less than 5% of the total number of AD cases and is mainly closely related to mutations in genes such as APP [7], presenilin 1 (PS1) and presenilin 2 (PS2). PS1 mutations, most of which are loss-of-function mutations that lead to decreased PS1 activity, are the main cause of FAD [8]. APP mutations change the processing of APP, leading to the production of neurotoxic Aβ42 and reduction in Aβ40. By triggering a variety of pathological mechanisms, Aβ42 selectively promotes the apoptosis or death of some nerve cells, which ultimately leads to the occurrence of AD. The PS proteins, encoded by the PS1 and PS2 genes, are an important part of gamma-secretase. Mutations in the PS1 gene result in the deletion of the hydrophilic loop domain of the encoded protein, which leads to a change in its conformation, which in turn affects the activity of γ-secretase and increases the production of Aβ42. In large cohort studies, most PS1 mutations were found to be missense mutations (Val97Leu and Ala136Gly), and a Val97Leu transgenic mouse model was established, in which the pathogenicity of this mutation was initially confirmed. Therefore, studying the pathogenesis of PS1 mutations in FAD is helpful for the understanding of the etiology of FAD and, at the same time, has guiding significant understanding of the etiology of sporadic AD. In addition, gene expression in the hippocampus of 76 AD patients and 40 healthy controls was analyzed. It was found that 12 genes are related to the course of AD, which may be used as markers for early diagnosis [9].

## 2. APP Structure and Metabolic Processes

The APP gene is located on human chromosome 21 and has 11 isoforms, which generally contain 305–770 amino acids, the most common of which are APP695, APP751 and APP770. APP695 is mainly highly expressed in neurons and lacks exons 7 and 8 compared to APP770. APP is a type I transmembrane protein containing an extremely large extracellular N-terminus, a single transmembrane region and a small cytoplasmic tail [10]. APP contains a heparin-binding structural domain (HBD), copper ion-binding structural domain (CuBD), acidic amino acid-enriched region (ACIDIC), Kunitz-type protease inhibitor (KPI) structural domain, central structural domain (CAPPD), RC structural domain and APP intracellular domain (AICD) (Figure 1A). Preliminary studies in our laboratory found that when the structural domain of ACIDIC was deleted, the Aβ product was significantly reduced, and as Aβ aggregation is the main cause of AD, this structural domain pattern needs to be studied in depth. The search for a model to restore β secretion and thus drug development for AD is ongoing.

APP metabolism mainly occurs via the amyloid and nonamyloid pathways [11,12,13]. The amyloid pathway is involved in the cleavage of APP by β-secretase into sAPPβ and a C-terminal fragment of 99 amino acids (C99); C99 is further cleaved by γ-secretase into the AICD and Aβ, and it is worth noting that this pathway does not produce over 90% of the total Aβ and this process is the only pathway that produces Aβ. It is worth noting that different modifications of APP affect the production of Aβ. There are two main types of Aβ, namely, Aβ40 and Aβ42. Aβ40 is relatively high in content, but Aβ42 is more neurotoxic due to its strong hydrophobicity and greater likelihood of aggregating. The nonamyloid pathway is involved in the cleavage of APP by α-secretase into sAPPβsAPPα and a C-terminal fragment containing 83 amino acids (C83); C83 is further cleaved by γ-secretase into the AICD and P3, and these small fragments are cleared by neurons (Figure 1B).

The normal metabolism of APP produces a soluble fragment that is nutritive to the cell, and only mutations in the gene make it more susceptible to cleavage by the corresponding secretase. There are many mutations in APP, mostly on either side or in the middle of the Aβ, and they can affect the pathological process of AD by different mechanisms [14]. Mutations in the N-terminus of Aβ increase the Aβ product and may be a more suitable recognition site for γ-secretase [15]. Mutations in the C-terminus of Aβ significantly increase the proportion of Aβ42, which easily aggregates and is then deposited by fibrosis, leading to the formation of diffuse age spots. Aβ can also cause oxidative stress and calcium inward flow, which can damage mitochondria, lead to neuronal dysfunction and activate apoptosis-related proteins to initiate the apoptotic program. In addition, Aβ can lead to neuronal apoptosis by causing an inflammatory response and neurofibrillary tangles in the brain, which is a pathological feature of AD and an important cause of its formation and development [16,17].

APP is a secreted protein; that is, the protein is synthesized in the cell and secreted to act outside of the cell. After initiation of translation on free ribosomes, it is transferred via signal peptides to the endoplasmic reticulum, where it continues to be synthesized and modified by N-terminal glycosylation processing in the lumen of the endoplasmic reticulum. The majority of immature APP is transported via vesicles to the Golgi apparatus, where it undergoes chemical modification processes, such as phosphorylation and C-terminal glycosylation, to become an active protein. Mature APP is partially processed in the Golgi by active ADAM10 enzymes [18]. However, the distribution of ADAM10 in the Golgi apparatus is relatively small relative to the cytoplasmic membrane; thus, processing is less efficient, and small amounts of BACE1 are also present in this organelle. Most of the mature APP is secreted in vesicles via lattice clathrin-mediated endocytosis to the cytoplasmic membrane, where it is enzymatically processed by α-secretase. In addition, APP secreted to the cell surface is transported by endosomes into various intracellular organelles within a short period of time, a process that is mainly mediated by one of APP’s own structural domains, that is, the endocytic module (YENPTY) [19,20]. The endosomes and lysosomes also contain γ-secretase, so APP passes through the plasma membrane to the endosomes where it is processed by β-secretase and γ-secretase to produce Aβ, which is degraded in the lysosomes.

In summary, APP is synthesized and secreted extracellularly from the endoplasmic reticulum to the Golgi apparatus, where it becomes a mature protein and is transported via the endocytic transport pathway to other intracellular organelles or structures for processing. By immunohistochemistry, APP can be observed in the endoplasmic reticulum, Golgi, endosome, lysosome and cell membrane.

## 3. α-Secretase

α-Secretase activity is derived from the “a disintegrin and metalloprotease” (ADAM) family. This family plays an important role in mediating signal transduction, cell membrane surface proteolysis and angiogenesis [21,22]. ADAM proteins contain different domains, and the original domain is removed when they enter the Golgi apparatus to perform different functions. The three most common subtypes in the ADAM family are ADAM9, ADAM10 and ADAM17, among which ADAM10 plays a major role in APP metabolism [23]. The activity of α-secretase is mainly on the cell membrane, and its activity may usually involve the combined effects of several proteins. For example, phorbol ester can activate protein kinase C in vivo and in vitro, thereby enhancing the activity of α-secretase and leading to increased secretion of sAPPα [24]. The distribution and protease activity of ADAM10 is regulated through synaptic activity, and ADAM10 enters the cell to be endocytosed by the action of the lattice clathrin adaptor protein AP2. In addition, it was found that overexpression of SIRT1 in APPswe/PSEN1dE9 transgenic mice reduced amyloid plaques in the mouse brain. Retinoic acid deacetylates the retinoic acid receptor in a SIRT1-dependent manner, which activates the gene expression of ADAM10 and biases the APP metabolic pathway toward the nonamyloid pathway, producing less Aβ.

In addition to acting on APP, ADAM proteins can catalyze Notch receptors and regulate the normal development of the body [22]. APP-mediated ADAM mediates the hydrolysis of the Notch N-terminus, and Notch is further cleaved by γ-secretase to produce an active form of the Notch intracellular domain (NICD). Notch receptors are located on the surface of the cell membrane and function in signal transduction from the surface of the cell membrane to the cell. Most importantly, ADAM17, with adhesion and proteolytic properties, is a metalloprotease involved in the shedding of extracellular domains of various proteins. The shedding of extracellular domains of proteins can affect a variety of structural and functional molecules; for example, it can convert tumor necrosis factor-α (TNF-α) into soluble TNF-α. In addition to cleaving TNF-α, ADAM17 also plays a role in the processing of many other protein substrates, such as cell adhesion molecules, cytokines, growth factor receptors, and epidermal growth factor receptors (EGFRs). Approximately 2–4% of the protein on the cell surface must shed its extracellular domain to perform normal physiological functions. For APP, ADAM17-mediated shedding of the extracellular domain of proteins does not seem to require a specific sequence. The minimum requirement is that the substrate is in an α-helical conformation and the hydrolyzed peptide bond needs to be a certain distance away from the cell membrane [25]. The shedding of the extracellular domain of the protein is the first step in the two consecutive cleavage events of the regulated intramembrane proteolysis (RIP) substrate, and the second step occurs inside the hydrophobic cell membrane molecule.

## 4. β-Secretase

The main β-secretase expressed in the brain is “β-site APP-cleaving enzyme-1” (BACE1) [26]. It is also located on the cell membrane. First, the immature protein is synthesized in the endoplasmic reticulum in an inactive form, and then it is transported to the Golgi apparatus to become active BACE1 [27]. Active BACE1 is transported to the plasma membrane through secretory vesicles and mainly accumulates in the lipid raft area. Although the vast majority of BACE1 in cells exists in the form of integral membrane proteins, a small part of BACE1 can still fall off the extracellular domain. BACE1 in the extracellular space and lipid rafts is endocytosed into the endosome to enzymatically process APP or is transported to the lysosome to be degraded. APP and BACE1 enter the cell through different endocytic pathways, where BACE1 enters the early endosome from outside the cell by means of ARF6, while APP enters the cell through clathrin-mediated endocytosis [28,29]. Finally, the two meet in Rab5-positive endosomes, and APP is processed by BACE1 secretase [30]. Of note, sAPPα, one of the products formed by α-secretase processing of APP, can actually inhibit BACE1 enzyme activity [31].

Generally, BACE1 secretase has higher enzymatic activity in acidic environments. There are two very important sites on this domain, namely, the DTGS and DSGT sites, which enable BACE1 to exert its enzymatic activity. The DTGS amino acid sequence is between site 93 and 96, and the DSGT sequence is between site 289 and 292. These two important sites enable BACE1 to process APP in the correct topological direction [32]. In the diagnosis of AD, the expression of BACE1 can also be used as one of the diagnostic methods [33,34,35]. According to research, the expression of BACE1 in the brains of most AD patients is significantly higher than that in the brains of normal people. In addition, there is a large accumulation of BACE1 around the nerve synaptic terminals of AD patients, which indicates that the abnormal accumulation of BACE1 at the presynaptic terminal contributes to the pathogenesis of AD [36].

## 5. γ-Secretase

γ-Secretase exists in multicellular animals. Studies on γ-secretase show that the secretase includes at least four transmembrane proteins: presenilin (PS), presenilin-enhancer 2 (PEN2), nicastrin (NCT), and anteriorpharynx-defective-1 (APH-1) [37,38], so it is a multisubunit high-molecular-weight complex. As γ-secretase is composed of a variety of proteins, the different combinations make γ-secretase highly heterogeneous, and it will perform different functions in different cells, tissues, organs, and at different times. PS is a multitransmembrane protein whose intracellular segment is hydrolyzed to form a heterodimer including the N-terminus and C-terminus, each providing an Asn residue, which is an essential component of the structure of γ-secretase. PEN2 can hydrolyze PS, thereby activating γ-secretase [39,40]. NTC is a type I transmembrane protein. Two large and small groups in the extracellular region are connected by a hydrophobic structure. Among them, the large group interacts with the extracellular matrix and can help γ-secretase bind to its receptor. During the assembly process of the γ-secretase complex, APH-1 first interacts with NCT to form a stable complex and then binds to the transmembrane end of PS. Finally, NTC undergoes glycosylation modification in the Golgi apparatus to form a mature γ-secretase complex. Less than 10% of mature γ-secretase is transported to the plasma membrane and stored in lipid rafts, while the rest is transported intracellularly [41,42].

APP was the first substrate discovered that was cleaved by γ-secretase, but studies have found that γ-secretase plays an important role in the hydrolysis of a large number of type I transmembrane proteins [43]. In the processing of APP, intracellular fragments processed by α-secretase or β-secretase can be processed by γ-secretase into P3 and AICD fragments. γ-Secretase activity can also be detected in lysosomes and endosomes. It is especially worth mentioning that the optimal pH for BACE1 activity is 5.5. Therefore, lysosomes and late endosomes are the main places where Aβ is produced.

## 6. Alternative Secretases: Matriptase-2 and MT5-MMP

Matriptase-2, also known as TMPRSS6, is a member of type II transmembrane serine proteases [44,45]. Matriptase-2 consists of a short intracytoplasmic amino terminus, a transmembrane region, and a main body region of the extracellular segment [45]. Matriptase-2 is mainly expressed in the liver. In addition, it is expressed in small amounts in tissues such as the kidney, lung, brain and uterus [46,47].

As most trypsins, matriptase-2 is also synthesized in the form of an inactive zymogen [45], and its activation requires digestion of the conservative activation site R576-I577. The activated matriptase-2 is still connected to the main body area, bound to the membrane through a disulfide bond [46]. To date, the relevant mechanisms regulating the synthesis and activation of matriptase-2 are still not very clear. The endocytosis and shedding of extracellular segments of matriptase-2 have also been reported, which may represent another mechanism for regulating the expression and activation of matriptase-2 [48,49]. Matriptase-2 plays an important role in many pathological and physiological processes in mammals, and its abnormal function can lead to cancer, iron deficiency anemia, and cardiovascular disease. Meanwhile, studies have found that increased expression of metalloprotease meprin β is associated with AD. Therefore, regulating the activity of meprin β may be a suitable strategy for the treatment of this condition. The activation of meprin β requires proteolytic maturation by serine proteases. Matriptase-2, as a new and effective activator, can activate meprin β on the cell surface to shed the extracellular domain of APP, thereby reducing Aβ deposition [50].

Membrane-type 5 matrix metalloproteinase (MT5-MMP) belongs to the multigene family of Zn^2+^ MMPs, which are widely involved in different physiological and pathological processes [51]. MT5-MMP is mainly expressed in the nervous system, mainly in neurons and, to a lesser extent, in astrocytes, microglia and endothelial cells [52,53]. MT5-MMP is a 645-amino acid transmembrane glycosylated proteinase that is intracellularly activated by the Ca^2+^-dependent proprotein convertase furin. MT5-MMP has been detected in the brain amyloid plaques of patients with AD [54]. Compared with wild-type mice, TgMT5^−/−^ mice, lacking MT5-MMP, show decreased Aβ load and neuroinflammation and improved spatial memory in the early pathology. MT5-MMP appears to be a protease with the ability to promote amyloid production, which can reduce amyloidosis/neuroinflammation and related functional defects [55].

## 7. Shedding of the Extracellular Domain of the Membrane Protein APP

The production of Aβ peptide through the hydrolysis and processing of APP by β- and γ-secretase is the core of the etiology of AD. APP is a membrane protein. The shedding of the extracellular domain of membrane proteins is a posttranslational modification [56]. Recent studies have shown that some membrane proteins that shed their extracellular domain can be used as potential drug targets and biomarkers for diseases. The protease cleaves the membrane protein substrate near or within its transmembrane region and produces a soluble extracellular domain during the shedding process [57]. Some shedding enzymes are called secretases because fragments of the substrate that are cleaved may be secreted [58].

The human genome contains approximately 600 protease-encoding genes [59], some of which have many corresponding substrates, and some have only one substrate identified thus far. Among them, the typical shedding enzymes for the substrate APP are ADAM10 and BACE1, and cleavage of the substrate APP by ADAM10 occurs under normal physiological conditions. It is worth noting that ADAM10 does not act as the α-secretase of APP, thereby preventing the production of neurotoxic Aβ, so it is considered a drug target for AD [60]. The metalloprotease ADAM10 is important for Notch pathway-dependent brain development. In ADAM10 knockout mice, sAPPα production was reduced, the expression of sAPPβ and endogenous Aβ was increased, and the clinical phenotypic neuromotor ability and learning ability were decreased [61]. Another exfoliating enzyme for the substrate APP is BACE1 [62]. BACE1 is the main β-secretase of APP, and the cutting and processing of APP leads to the production of Aβ. Studies have found that BACE1 not only produces Aβ protein but also destroys the factors required for cells to produce PKA. PKA plays an important role in memory production, so BACE1 is a promising drug target for the treatment of AD. The experiment found that long-term inhibition of BACE1 affects the formation of synapses in mice. It is assumed that the inhibitor changed the metabolism of BACE1 substrates [63].

## 8. Conclusions and Outlook

The number of people with dementia is particularly high in China and poses a great challenge to the country’s economic development. Therefore, it is especially important to study the mechanisms of AD, which are currently mainly thought to be the deposition of Aβ in the brain, the formation of neuroinflammatory patches and the loss of neurons. The study of APP and its metabolites is an important part of the investigation into the mechanism of AD, and there are many potential drug targets in the pathogenesis, for which it is scientifically important and clinically promising to design advanced drugs for the treatment of AD.

Under normal conditions, APP only produces soluble fragments that are nutritive to the cell; when mutations occur in the gene, the associated secretase action of APP is likely to increase. APP has two classical metabolic pathways, and α-secretase, β-secretase and γ-secretase are all widely considered to be sheddases. The extracellular domain shedding of membrane proteins that involves sheddases is normally controlled by signal transduction, and when deregulated, extracellular domain shedding is associated with pathologies such as inflammation and AD. However, sheddases and substrate proteins do not function in a one-to-one manner, and the mode of action is unclear. To better understand the molecular mechanisms underlying the substrate specificity of sheddases, it is necessary to assign individual substrates to sheddases and to determine how protein hydrolytic cleavage affects substrate function. With further research into the pathogenesis of AD and the refinement of various potential drug targets, it is believed that appropriate drugs can be identified to alleviate the suffering of AD patients.

## Figures and Tables

**Figure 1 membranes-11-00983-f001:**
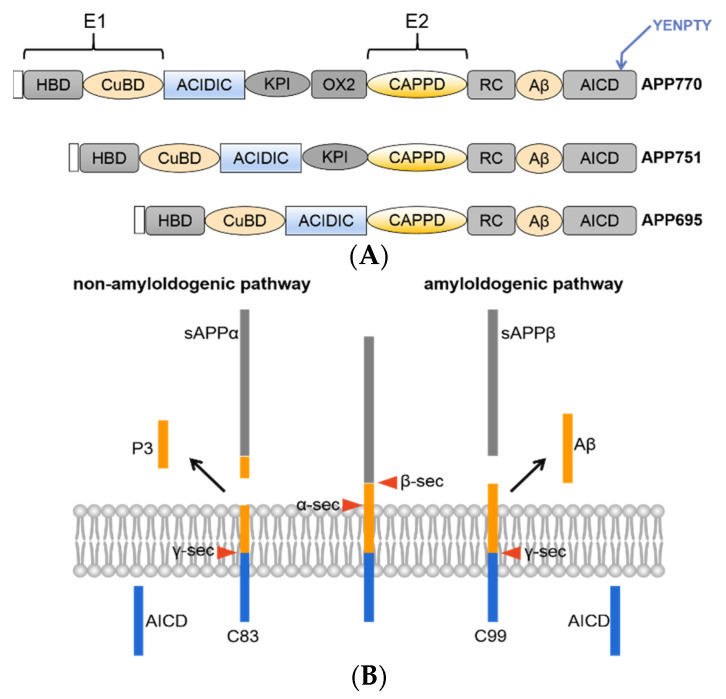
(**A**) Schematic overview of the domain structures of APP family proteins. All APP family members share conserved E1 and E2 extracellular domains, an acidic domain (Ac) and the YENPTY motif at the carboxyl terminus. (**B**) Schematic diagram of APP processing. There are two classical peptide-generating pathways, namely, the amyloid pathway and nonamyloid pathway.

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
