# Peer review of "Secretases Related to Amyloid Precursor Protein Processing"

_membranes, 2021, doi:10.3390/membranes11120983_

Round 1

Reviewer 1 Report

The study of the metabolism of amyloid precursor protein (APP) in the pathogenesis of Alzheimer's Diseases (AD) could provide a new direction for AD therapy. The review entitled "Secretases related to amyloid precursor protein processing" concisely summerizes the key role of secretases in the APP processing.

My major comments are:

KEYWORDS: the authors should add "APP processing" in the keywords

INTRODUCTION: the authors should add to the text the most important pathogenetic tracts of AD. AD is characterized by loss of neuronal brain mass and presence of amyloid plaque. Accumulation of amyloid ß protein is one of the earliest event in the disease process, occurring 10-20 years prior the onset of clinical symptoms.  Therefore several therapies have been aimed at reducing amyloid plaques, but they have not great results. For this reasons, it is important to focus on APP processing.

Author Response

Answers to reviewer 1
The study of the metabolism of amyloid precursor protein (APP) in the pathogenesis of Alzheimer's Diseases (AD) could provide a new direction for AD therapy. The review entitled "Secretases related to amyloid precursor protein processing" concisely summerizes the key role of secretases in the APP processing.

Major comments

1.KEYWORDS: the authors should add "APP processing" in the keywords.

Answer: Thank you for your suggestion. The keyword "APP processing" has been added to the manuscript.

2.INTRODUCTION: the authors should add to the text the most important pathogenetic tracts of AD. AD is characterized by loss of neuronal brain mass and presence of amyloid plaque. Accumulation of amyloid ß protein is one of the earliest event in the disease process, occurring 10-20 years prior the onset of clinical symptoms. Therefore
several therapies have been aimed at reducing amyloid plaques, but they have not great results. For this reasons, it is important to focus on APP processing.

Answer: Thank you for your suggestion. I have added the pathogenesis of AD in line 26.

Reviewer 2 Report

This review manuscript describes the molecular mechanisms underlaying the processing of APP and other substrates, especially focuses on proteases, called secretase. The manuscript is somehow summarized well, however, there are many points to be correct. In addition, I think the authors should include more citations. This manuscript has too few references.

Line 11, what is the aspartic protein? I have never heard before for APP.

Line 38, there is a description for preliminary studies. I think it is inadequate for a review paper.

Line 46, authors described "and it is worth noting that this pathway does not produce over 90% of the total Aβ." I cannot get what the authors intended to describe with this description. APP is cleaved into Abeta through the amyloidgenic pathway and this process is the only pathway to produce Abeta. Therefore, this pathway produced nearly 100% of Abeta.

Line 50, sAPPβ should be sAPPalpha.

Too few references are cited throughout the manuscript. For example, not only these but also throughout this manuscript, lines 84 to 90, positions indicated with [] requires proper references: The majority of immature APP is transported via vesicles to the Golgi apparatus, where it undergoes chemical modification processes such as phosphorylation and C-terminal glycosylation [] to become an active protein. Mature APP is partially processed in the Golgi by active ADAM10 enzymes []. However, the distribution of ADAM10 in the Golgi apparatus is relatively small relative to the cytoplasmic membrane []; thus, pro-cessing is less efficient [], and small amounts of BACE1 are also present in this organelle []. Most of the mature APP is secreted by vesicles through the lattice protein to the cyto-plasmic membrane [], where it is enzymatically processed by α-secretase [].

Lines 90 and 115, what is the lattice protein?

Line 111, protease C should be protein kinase C.

Line 120, ADAM should be ADAMs or ADAM proteins.

Line 121, APP mediates... should be ADAM mediates...

Line 137, describe what RIP abbreviates.

Lines 186 to 188, a description, "In the processing of APP, intracellular fragments 186 processed by α-secretase or β-secretase can be processed by γ-secretase into P3 and AICD 187 fragments." is wrong. P3 is derived from by alpha-secretase, not by beta-secretase.

Lines 205 to 206, a description, "It is worth noting that ADAM10 does not act as the 205 α-secretase of APP" is wrong. ADAM10 acts as alpha-secretase.

Author Response

This review manuscript describes the molecular mechanisms underlaying the processing of APP and other substrates, especially focuses on proteases, called
secretase. The manuscript is somehow summarized well, however, there are many points to be correct. In addition, I think the authors should include more citations. This manuscript has too few references.

Major comments

1. Line 11, what is the aspartic protein? I have never heard before for APP.

Answer: Thank you for your suggestion. I accidentally added an extra word when I wrote it
,and I have deleted it in the text. What I want to express is: APP is a single
transmembrane protein.

2. Line 38, there is a description for preliminary studies. I think it is inadequate for a review paper..

Answer: Thank you for your suggestion. Supplements have been made to this description.

3. Line 46, authors described "and it is worth noting that this pathway does not produce over 90% of the total Aβ." I cannot get what the authors intended to describe with this
description. APP is cleaved into Abeta through the amyloidgenic pathway and this process is the only pathway to produce Abeta. Therefore, this pathway produced nearly
100% of Abeta.

Answer: Thank you for your suggestion. The writing was not cautious enough, the text has been changed to: "and this process is the only pathway to produce A
β, it is worth
noting that different modifications to APP will affect the production of A
β" .

4. Line 50, sAPPβ should be sAPPalpha.

Answer: Thank you for your suggestion. SAPPβ has been changed to sAPPα.

5. Too few references are cited throughout the manuscript. For example, not only these
but also throughout this manuscript, lines 84 to 90, positions indicated with [] requires proper references: The majority of immature APP is transported via vesicles to the Golgi apparatus, where it undergoes chemical modification processes such as phosphorylation and C-terminal glycosylation [] to become an active protein. Mature
APP is partially processed in the Golgi by active ADAM10 enzymes []. However, the distribution of ADAM10 in the Golgi apparatus is relatively small relative to the
cytoplasmic membrane [].

Answer: Thank you for your suggestion. References have been added at appropriate places in the article.

6. Lines 90 and 115, what is the lattice protein?

Answer: Thank you for your suggestion. What should be described in the manuscript is clathrin, and the translation is not cautious enough. The text has been corrected to "clathrin protein".

7. Line 111, protease C should be protein kinase C.

Answer: Thank you for your suggestion. The text has been corrected to "protein kinase C".

8. Line 120, ADAM should be ADAMs or ADAM proteins.

Answer: Thank you for your suggestion. The text has been corrected to "ADAM proteins".

9. Line 121, APP mediates... should be ADAM mediates...

Answer: Thank you for your suggestion. The text has been corrected to "ADAM mediates".

10. Line 137, describe RIP abbreviates.what

Answer: Thank you for your suggestion. The abbreviation RIP in the text is regulated intramembrane proteolysis, which has been added in the text.

11. Lines 186 to 188, a description, "In the processing of APP, intracellular fragments 186 processed by α-secretase or β-secretase can be processed by γ-secretase into P3 and AICD 187 fragments." is wrong. P3 is derived from by alpha-secretase, not by beta-secretase.

Answer: Thank you for your suggestion. Changed in the text to "In the processing of APP, intracellular fragments 186 processed by
α-secretase can be processed by γ-secretase into P3 and AICD 187 fragments".

12. Lines 205 to 206, a description, "It is worth noting that ADAM10 does not act as the 205 α-secretase of APP" is wrong. ADAM10 acts as alpha-secretase.

Answer: Thank you for your suggestion. Changed in the text to "It is worth noting that ADAM10 does act as the α-secretase of APP".

Round 2

Reviewer 2 Report

I have no additional comment, therefore the manuscript can be accept as it is.